# Recent Advances in Fluorescent Methods for Polyamine Detection and the Polyamine Suppressing Strategy in Tumor Treatment

**DOI:** 10.3390/bios12080633

**Published:** 2022-08-12

**Authors:** Bingli Lu, Lingyun Wang, Xueguang Ran, Hao Tang, Derong Cao

**Affiliations:** 1Key Laboratory of Functional Molecular Engineering of Guangdong Province, School of Chemistry and Chemical Engineering, South China University of Technology, 381 Wushan Road, Guangzhou 510641, China; 2Institute of Animal Science, Guangdong Academy of Agricultural Sciences, Ministry of Agriculture Key Laboratory of Animal Nutrition and Feed Science in South China, State Key Laboratory of Livestock and Poultry Breeding, Guangzhou 510641, China

**Keywords:** polyamines, detection, suppressor strategies, tumor

## Abstract

The biogenic aliphatic polyamines (spermine, spermidine, and putrescine) are responsible for numerous cell functions, including cell proliferation, the stabilization of nucleic acid conformations, cell division, homeostasis, gene expression, and protein synthesis in living organisms. The change of polyamine concentrations in the urine or blood is usually related to the presence of malignant tumors and is regarded as a biomarker for the early diagnosis of cancer. Therefore, the detection of polyamine levels in physiological fluids can provide valuable information in terms of cancer diagnosis and in monitoring therapeutic effects. In this review, we summarize the recent advances in fluorescent methods for polyamine detection (supramolecular fluorescent sensing systems, fluorescent probes based on the chromophore reaction, fluorescent small molecules, and fluorescent nanoparticles). In addition, tumor polyamine-suppressing strategies (such as polyamine conjugate, polyamine analogs, combinations that target multiple components, spermine-responsive supramolecular chemotherapy, a combination of polyamine consumption and photodynamic therapy, etc.) are highlighted. We hope that this review promotes the development of more efficient polyamine detection methods and provides a comprehensive understanding of polyamine-based tumor suppressor strategies.

## 1. Introduction

Polyamines (spermine, spermidine, and putrescine) consisting of two or more primary amino groups are aliphatic organic cationic compounds that are widely found in organisms (Figure 1) [1]. The homologs of polyamines (such as cadaverine, norspemine, homospemidine, and thermospermine) at lower concentrations play a less significant role [2]. The total number of polyamines found in living systems is on the millimole level, but the concentration of free polyamines is relatively even lower because cationic polyamines are usually bound to anionic nucleic acids, proteins or phospholipids, DNA, RNA, proteins, etc. [3].

Polyamines at appropriate concentrations play several positive roles in many metabolic processes [4]. Under physiological conditions, polyamines can prevent the denaturation of DNA and stabilize the structure of nucleic acid [5]. Moreover, they participate in catalysis and the controlled biosynthesis of nucleic acids [6]. Some results indicate that the synthesis of DNA is stimulated by spermidine [7]. In addition, polyamines are responsible for cell growth, differentiation, gene regulation, cell division, homeostasis, ion channel function, protein synthesis, cell cycle progression, and other biological functions [8,9]. The level of polyamine concentrations is regulated by several key enzymes of synthesis and catabolism [10,11].

However, an excess accumulation of polyamines may lead to apoptosis and various allergic disorders [12]. Some researchers reveal that the rapid proliferation and differentiation of tumor cells are dependent on the intracellular polyamines’ concentrations [13,14]. Due to their high demand for polyamines, tumor cells have to uptake a large number of polyamines from the extracellular matrix to maintain their rapid growth, resulting in a much higher level of polyamines in tumor cells than is found in normal cells [15]. The levels of putrescine, spermidine, and spermine in blood samples in cancer patients are significantly higher than those found in healthy people [16]. Therefore, a high level of polyamines is regarded as a biomarker for the timely diagnosis of cancer in certain tumors [17]. For this reason, it is important to detect and monitor non-bonded intracellular polyamines.

Since the regulation of biosynthesis and catabolism of polyamines is particularly important, there is an urgent need to develop chemotherapeutic drugs to inhibit tumor proliferation, differentiation, or metastasis [18]. For example, the biological pathways for inhibiting polyamine biosynthesis, activating polyamine catabolism, and blocking polyamine transport have been investigated. Likewise, new anti-cancer therapies, such as supramolecular chemotherapy, using polyamine conjugates to interfere with DNA function, and irreversible chemical reactions that consume polyamines have received scientific attention.

Fluorescence-sensing technology offers numerous outstanding advantages, such as high sensitivity, simple operation, a fast response speed, low cost, real-time visualization, and non-destructive monitoring. This review summarizes the most recent advances in polyamines detection based on the fluorescence method (a supramolecular sensing system, chromophore reaction-based fluorescent probes, fluorescent small molecules, and fluorescent nanoparticles). The tumor suppressor strategy related to polyamines (such as polyamine conjugates, polyamine analogs, combinations that target multiple components, spermine-responsive supramolecular chemotherapy, a combination of polyamine consumption and photodynamic therapy, etc.) is highlighted (Figure 1). We believe that this review will provide a deep understanding and offer important guidance for the design and development of polyamine-based detection methods and antitumor drugs.

## 2. Polyamine Detection Methods

Numerous measuring methods and instruments of analysis, such as the enzyme-linked immunosorbent assay (ELISA) [19], a combination of mass spectrometry (MS) with gas chromatography (GC) or liquid chromatography (LC) [20], high-performance liquid chromatography (HPLC) [21], an antibody-dependent assay [22] (ADA), and nuclear magnetic resonance (NMR) [23] have been developed. However, some drawbacks such as low selectivity, expensive equipment costs, the need for professional staff, and complicated sample pre-treatment requirements are frequently encountered. On the other hand, fluorescence-sensing technology has attracted more and more attention in the field of polyamine detection, due to its high sensitivity, simple operation, fast response speed, low cost, real-time visualization, and non-destructive monitoring. Therefore, we focus on fluorescent methods for polyamine detection in this review. Figure 1 shows the roadmap for polyamine monitoring methods. Until now, small organic molecules [24,25,26,27,28], conjugated polymers [29,30,31,32], dye-assembled nanotubes [33], hydrogel hybrids [34], dye-embedded micelles [35,36], nanoparticles [37,38], nano-Au [39], or quantum dots [40] have been utilized to identify and detect polyamines. Most of them are involved in displacement or aggregation-based sensing mechanisms.

### 2.1. Supramolecular Sensing System for Polyamine Detection

Recently, spermine detection, based on a supramolecular strategy, has received more scientific attention due to the strong complexation ability between the macrocyclic host and spermine [41]. In some cases, the modulation of the supramolecular system’s morphologies by spermine yielded obvious fluorescence signals [42]. By the use of the multi-cationic character of tetracationic spermine (p*K*_a_ = 11.50, 10.95, 9.79, 8.90) and tricationic spermidine (p*K*_a_ = 11.56, 10.80, 9.52), the generation of emissive excimers was investigated for the selective detection of spermine and spermidine [43]. Besides these strategies, the competitive binding induced by spermine is a widely utilized sensing mechanism [44,45,46,47].

The water-soluble cucurbit[7]uril (CB[7]) was selected as a popular host because of its strong binding ability toward polyamines. The highly emissive “dumbbell-shape” supraamphiphiles, based on perylene diimides (compound **1**) and CB[7], were utilized as a supramolecular sensor for the fast and ultra-sensitive detection of spermine [44]. Due to its high affinity with CB[7], spermine can preferably bind to CB[7] (binding constant K = 2.6 × 10^7^ M^−1^), leading to the dissociation of the supra-amphiphiles and generating fluorescence quenching. The supraamphiphiles showed good selectivity for spermine over other structural analogs (spermidine, putrescine, L-arginine, L-lysine, etc.). This high sensitivity was maintained even in the presence of a low concentration of spermine (about 10 nM) (Figure 2).

The cationic pyridyl functionalized compound **2** was an aggregation-induced-emissive (AIE) active dye, which was non-emissive in solution. Upon complexation with CB[7] via host–guest interaction, a fluorescent supramolecular system was formed [45]. The binding constant between **2** and CB[7] was found to be 3.77 × 10^4^, 2.22 × 10^4,^ and 2.86 × 10^4^ M^−1^ at a pH of 3.0, 7.0, and 10.0, respectively. Since the binding constant (2.6 × 10^7^ M^−1^) between spermine and CB[7] was 1000-fold higher than **2**-CB[7], **2** was released from the supramolecular system. The fluorescence at 537 nm was turned off. The detection limit of spermine was found to be 1.0 μM (Figure 3).

Bhosle et al. developed a three-component supramolecular sensing system containing a fluorescent compound, **3**, CB[7], and hydroxyapatite nanoparticles (HAp NPs), which can be used to detect polyamines with high sensitivity [46]. In this case, compound **3** spontaneously interacted with HAp NPs by electrostatic interaction and with CB[7] via host–guest interaction. As a result, a three-component supramolecular system showed strong emissions at 615 nm (Figure 4). Spermine showed a higher binding ability to CB[7] than compound **3**, leading to a higher binding constant (2.6 × 10^7^ M^−1^ for spermine-CB[7] vs. 5.6 × 10^4^ M^−1^ for **3**-CB[7]). Upon the addition of polyamines, compound **3** was displaced from CB[7] and quenching fluorescence was observed. The low limits of detection of 1.4, 3.3, and 17.2 ppb for spermine, spermidine, and cadaverine were obtained, respectively.

Similarly, a three-component supramolecular sensing assembly based on the water-soluble TPE derivative (**4**), hydroxyl cucurbit[6]uril (CB[6]OH), and hydroxyapatite nanoparticles (HAp NPs) showed excellent sensing performance for the detection of spermine and spermidine in human urine and blood [47]. A binary conjugate via host−guest complexation between compound **4** and CB[6]OH) at a 1:4 ratio was formed. The further addition of HAp NPs to the binary conjugate led to a 13-fold enhancement by more effective aggregation. The corresponding three-component supramolecular assembly showed the fluorescence turn-off response at 474 nm to spermine and spermidine, but not to other amines. LOD values of 1.4 × 10^−8^ and 3.6 × 10^−8^ M for spermine and spermidine were reported, respectively. The much stronger affinity between spermine and CB[6]OH created compound **4** in monomeric form, leading to emission quenching (Figure 5).

The ionic self-assembly of the benzimidazolium platform and ciprofloxacin-Tb^3+^ complex were developed for selective spermine detection with the fluorescence quenching response [48,49]. Guo et al. designed a supramolecular assembly of an amphiphilic sulfocalix[5]arene (SC5A12C) with lucigenin (LCG) via host–guest interaction [50]. The fluorescence of LCG was quenched due to the strong binding capacity between SC5A12C and LCG. In the presence of overexpressed spermine in cancer cells, the fluorescence was fully recovered due to the competition complex between spermine and SC5A12C. Co-assembled folate further promoted the cellular uptake by folate receptor-overexpressing cancer cells.

Yang’s group reported polyamine detection, based on pyrene excimer fluorescence, through the synergistic/competitive complexation among pyrene compounds (**5**), polyamines, γ-cyclodextrins (γ-CD), and CB[7] [51]. In this case, γ-CD accommodated compound **5** with binding constants of 5 × 10^6^ M^−1^ for 1:2 binding modes, leading to the excimer emission of pyrene. Since a 1:1 host–guest complexation between **5** and CB[7] was identified, the addition of CB[7] yielded a decrease in the excimer fluorescence, accompanied by an increase in monomer fluorescence. On the other hand, polyamines can bind strongly to CB[7]. For example, spermine showed an association constant two orders higher of 1.28 × 10^6^ M^−1^ than that of pyrene. The addition of urinary polyamines to a three-component supramolecular system (**5**, γ-CD, and CB[7]) competitively captured CB[7] and formed a complex of γ-CD-**5** at 1:2 binding, enabling the recovery of the excimer fluorescence (Figure 6).

### 2.2. Polyamine Detection Based on Chromophore Reaction

The chromophore-reaction-based fluorescent probes possess great advantages, such as a high selectivity to analytes and synchronous colorimetric and fluorescence changes [52,53,54,55,56,57]. Recently, we reported a series of pyrrolopyrrole *aza*-BODIPY (PPAB)-based fluorescent probes for polyamine detection, wherein the chromophore reaction-sensing mechanism was involved [58,59,60,61]. For instance, three PPAB dyes (**4a**–**4c**) showed high selectivity and sensitivity toward polyamines by colorimetric changes from green to yellow and a fluorescent turn-on process [62]. There was a hypsochromic shift over 225 nm in the absorption maximum and a 12-fold fluorescence enhancement. The detection mechanism study revealed a B–N bond cleavage, and a transamination and hydrolysis reaction was involved that generated much smaller conjugated molecules. More interestingly, the limit of detection up to ppb level and the response time on the second timescale were demonstrated (Figure 7).

It is important to improve the reaction rate between fluorescent probes and polyamines. We designed and synthesized a lactam-fused *aza*-BODIPY (compound **7**), which contained fewer cleaved imine and B–N functional groups [63]. The presence of spermidine and spermine induced an obvious color change from pink to yellow (130 nm hypsochromic shift of the absorption peak) and a 99% fluorescence “turn-off” response within 1 minute. Other amines failed to yield the same optical phenomenon. The high pseudo-first-order rate constants (*k_obs_*) of 15.67 × 10^−3^ S^−1^ and 8.99 × 10^−3^ S^−1^ for spermine and spermidine were present, respectively. A similar proposed sensing mechanism was involved through the B–N bond cleavage and hydrolysis reaction, to yield much smaller conjugated molecules (Figure 8).

The *aza*-Michael addition reaction between α,β-unsaturated nitrile and primary amines can proceed smoothly under mild conditions. The pyrrolopyrrole cyanine dye (**8**) contained α,β-unsaturated nitrile as Michael acceptors, which was further appended with a withdrawing boron atom to enhance the *aza*-Michael addition reactivity [64]. Through this strategy, a highly efficient fluorescent polyamine probe was developed. Compound **8** showed 158-fold higher *k_obs_* with putrescine than compound **6a**. As an efficient chromophore reaction-based probe for polyamine detection, the synergistic *aza*-Michael addition, B–N detachment, and hydrolysis reaction were involved between compound **8** and polyamines. The resulting low-conjugated product generated synchronous colorimetric and fluorescence changes (Δλ_ab_ = 188 nm and Δλ_em_ = 151 nm). A limit of detection up to the 62.1 nM level for spermine was obtained (Figure 9).

### 2.3. Fluorescent Small Molecules for Polyamine Detection

Barros et al. synthesized a tetraphenylethylene (TPE) derivative (**9**) containing two carboxylic acid groups as a fluorescent probe for the detection of spermine and spermidine [65]. The multi-cationic spermine and spermidine easily formed a complex with compound **9** by electrostatic- and hydrogen-bonding interactions at the physiological pH. As a result, a distinct emission enhancement at 473 nm (60- and 80-fold increase for spermine and spermidine, respectively) was shown because the restricted intramolecular rotations of compound **9** played a positive role. Conversely, diamines such as ethylenediamine, diethylenetriamine, cadaverine, and putrescine failed to generate such a fluorescence response. The LOD values of 0.70 µM and 1.17 µM for spermine and spermidine, respectively, were achieved in urine (Figure 10).

Huang et al. synthesized an AIE-active compound, **10,** based on TPE and pentiptycene, which showed a fluorescence “turn-on” response in the presence of spermine [66]. The sensing mechanism was ascribed to aggregation-induced fluorescence enhancement via electrostatic pairing and hydrogen bonding. For instance, the fluorescence intensity of compound **10** gave a 75-fold fluorescence enhancement at 480 nm upon the addition of 40 μM spermine. The detection limit regarding spermine was found to be 0.3 μM. The practical application for spermine detection in artificial urine was demonstrated (Figure 10).

### 2.4. Fluorescent Nanoparticles for Polyamine Detection

A dual emissive nanoprobe was prepared from the combination of yellow emissive mercaptopropionic acid-capped CdTe quantum dots (YQDs) and blue emissive carbon dots (BCDs), wherein the former and the latter were employed as sensing and reference fluorophores, respectively [67]. Since spermine or spermidine could dramatically quench the fluorescence of YQDs (λ_em_ = 570 nm) but did not affect the emission of BCDs (λ_em_ = 450 nm), the presence of spermine or spermidine induced a decrease in the I_570_/I_450_ ratio. As a result, the green emission turned into a pink emission with the increase of spermine or spermidine, to yield a ratiometric signal. The low limits of detection were found to be 0.2 µM for spermine and 2.1 µM for spermidine, respectively. By using this sensing mechanism, a combinational logic-gate fluorescence sensor was achieved. 

Gluconate-stabilized AuNPs (Glu-AuNPs) possessed an anionic character, which was utilized as a nanosensor for selectively detecting spermine in urine samples [68]. In the presence of spermine, the absorption band at 518 nm gradually red-shifted to 618 nm due to aggregation, generating an obvious color change from pinkish-red to blue. The strong interaction between spermine and Glu-AuNPs was confirmed via its high binding constant (2.18 × 10^6^ M^−1^). The ratio of A_618_/A_518_ showed a good linear relationship with spermine concentration. The LOD value was calculated to be 0.25 μM. When Glu-AuNPs were fabricated into a colorimetric strip-based kit, the sensing of spermine in urine samples was achieved.

Two cobalt-based metal–organic frameworks (MOFs) were developed for polyamine detection, wherein a strong emission at about 332 nm was largely quenched by spermine, spermidine, or putrescine [69]. Since host–guest interactions between the polyamines and the MOFs were involved, the donor-acceptor electron transfer process from polyamines to MOFs generated photoinduced electron transfer (PET), leading to fluorescence quenching. Putrescine has higher HOMO energy levels than spermidine and putrescine (−4.81 vs. −4.96, −5.01 eV). The higher the energy difference between the HOMO energy levels of MOFs (−5.91 eV) and putrescine, the more effective the PET process. Thus, in terms of the lowest limit of detection (LOD) values (0.24 μM) for putrescine rather than the LOD for spermine, spermidine was shown.

## 3. Tumor Polyamine-Suppressing Strategy

Several tumor polyamine-suppressing strategies have been developed, as follows. (1) Ornithine decarboxylase (ODC) and S-adenosylmethionine decarboxylase 1 (AMD1) are important for polyamine synthesis [70]. The α-difluoromethylornithine (DFMO), which acts as an irreversible suicide inhibitor of ODC, has been used to prevent and treat multiple cancers, such as pancreatic cancer, gastric cancer, lung carcinoma, neuroblastoma, endometrial cancer, and osteosarcoma [71]. (2) Highly regulated catabolic pathways are utilized to control the intracellular polyamine pool. The modulation of the polyamine catabolic enzyme produces decreasing polyamine content and induces the generation of toxic compounds. (3) Some inhibitors targeting the polyamine transport system (PTS) can hinder polyamine import and antagonize polyamine uptake. (4) Synthetic polyamines, including polyamine analogs and polyamine conjugates, possess anticancer activity against tumor cells.

### 3.1. Small Molecules Target Polyamine Metabolism

#### 3.1.1. Polyamine Analogs 

As potential antitumor drugs, polyamine analogs have a similar structure to natural polyamines but cannot be functionally substituted for them. That means that polyamine analogs can compete with natural polyamines, can enter cells through polyamine transport channels, which down-regulates the activities of various polyamine synthases, and can promote polyamine catabolism. In this way, the intracellular content of natural polyamines is depleted by inhibiting biosynthesis and inducing catabolism. For example, propylenediamine-based polyamine analogs and bis(ethyl)spermine analogs (**11a**–**11f**, Figure 11) were synthesized to inhibit the ODC activity and growth of tumor cells [72,73].

#### 3.1.2. Polyamine Conjugates

Polyamine conjugates are usually composed of polyamine and a DNA intercalator. The former is responsible for driving the conjugates to be transported to cancer cells by PTS, competitively inhibiting the uptake of exogenous polyamines by tumor cells. The latter can effectively embed into DNA, destroying many DNA protein interactions and interfering with their function, resulting in cell death. DNA intercalators are capable of intercalating into nucleic acid base pairs. The classic intercalators include naphthalimide, flavonoids, naphthoquinone, anthraquinone, and chalcone.

Some naphthalimide-polyamine conjugates significantly inhibited tumor growth and metastasis [74,75,76]. Xie’s group [74] designed and synthesized a series of naphthalimide-polyamine conjugates (**12a**–**12d**, **13a**–**13c,** and **14a**–**14b**). They found that compound **13b** exhibited stronger antitumor activity against hepatoma cells than the other compounds. Moreover, its activity was significantly higher than that of commercial antitumor drugs (amonafil and cisplatin). The mechanism of action revealed that **13b** can regulate the key enzymes of polyamine metabolism with SSAT and PAO and down-regulate the concentrations of spermine, spermidine, and putrescine, thereby inhibiting the rapid growth of the tumor cells (Figure 12).

Wang’s group [77] reported a series of flavonoid polyamine conjugates (**15a**–**15f**) and evaluated their anti-tumor properties through in vitro and in vivo experiments (Figure 13). Overall, the series was moderately cytotoxic to tumor cells. However, only compound **15a** showed good selectivity between hepatocellular carcinoma cells (HCC) and normal hepatocytes. When **15a** was combined with aspirin, the anti-HCC activity was improved. The H22 liver tumor growth and lung metastasis were effectively inhibited. The combinational effect was found, wherein **15a** increased the expression of apoptosis-related proteins and aspirin further amplified this effect. Other flavonoid polyamine conjugates can also initiate autophagy and inhibit the occurrence of apoptosis [78,79].

A major enzyme involved in maintaining DNA topology, 1,4-naphthoquinone is capable of inhibiting DNA human topoisomerase II-α (topo2α) and has been widely used in many cancer-related biological target inhibitors [80]. Follmer’s group [81] constructed a series of 1,4-naphthoquinone-polyamine conjugates (**16a**–**16c**) to study their anticancer properties (Figure 14). The coupling of polyamines could enhance the antitumor activity of 1,4-naphthoquinone on tumor cells. Porphyrin-polyamine conjugates [82], anthraquinone-polyamine conjugates [83], and chalcone polyamine conjugates [84] have also been investigated for anti-tumor purposes.

### 3.2. Combinations Target Multiple Components

Due to compensatory mechanisms, the single agents have failed to obtain satisfactory clinical effects. For example, increased polyamine uptake from the surrounding microenvironment is involved if polyamine synthesis is limited. The increased polyamine catabolism leads to an increase in the activity of the ODC polyamine biosynthetic enzyme. When polyamines are depleted in cells, the ODC expression levels are increased. A combination strategy combining two or more agents is more promising for cancer chemoprevention because the agents work synergistically, with decreased adverse effects. For example, “polyamine blocking therapy (PBT)” by a combination of polyamine biosynthesis and transport inhibitors can lead to efficient polyamine depletion. The combination of DFMO and catabolism inducers has shown enhanced antitumor activity. 

#### 3.2.1. Spermine-Responsive Supramolecular Chemotherapy

Supramolecular chemotherapy is devoted to utilizing supramolecular approaches to reduce the cytotoxicity of chemotherapeutic drugs and enhance their anticancer activity. The use of macrocyclic host molecules can encapsulate clinical anticancer drugs and reduce their toxicity to normal cells. Upon entering the polyamine multi-expression environment, the polyamine can competitively combine with the macrocyclic host, thereby causing the release of anti-tumor drug molecules.

Zhang and co-workers developed supramolecular chemotherapy based on cucurbit[n]uril and an antitumor drug [85,86,87]. Upon competitive replacement by spermine, the recovery of the antitumor activities of drugs was obtained. For example, the host−guest complex of dimethyl viologen (MV)-CB[7] was added to tumor cells with overexpressed spermine; subsequently, the antitumor activity of MV can be recovered by the competitive replacement of spermine [85]. When another clinical drug, oxaliplatin, was used as a model antitumor agent, the oxaliplatin-CB[7] complex exhibited more cooperatively enhanced antitumor activity than oxaliplatin itself (Figure 15) [86]. The cooperatively enhanced spermine consumption in tumor environments and the release of oxaliplatin were possible reasons.

The supramolecular polymeric chemotherapy results were also promising. PEG-functionalized CB[7] can encapsulate oxaliplatin to form a supramolecular polymeric complex [87]. Due to the presence of the PEG chain, the improved circulation performance of oxaliplatin was achieved. This combined effect led to enhanced cytotoxicity for tumor cells and decreased cytotoxicity for normal cells. 

Supramolecular chitosan nanogels (SCNs) were fabricated, based on phenylalanine-grafted chitosan and cucurbit[8]uril (CB[8]) by stimuli-responsive host−guest interactions [88]. Doxorubicin hydrochloride (DOX), a chemotherapeutic agent, was entrapped in the matrix to yield DOX-SCNs, with an excellent drug loading efficiency. In the presence of spermine, the encapsulated DOX was selectively released due to a strong host–guest interaction between CB[8] and spermine. The SCNs were efficiently internalized by the cells, and the DOX-SCNs exhibited specific, potent activity against spermine-overexpressed A549 cancerous cells.

Since ZnO can interact with DOX and CB[7], Chen et al. constructed a smart supramolecular cargo of ZnO-DOX-CB[7] via the ion-dipole interaction [89]. Due to the higher binding affinity between CB[7] and spermine, the nanocomplex could release DOX and enhance antitumor activity through competitive displacement in tumor microenvironments. Similarly, a heptaplatin-CB[7] supramolecular chemotherapy platform was constructed via host−guest interaction for killing tumor cells [90], since overexpressed spermine competitively exchanged heptaplatin from the supramolecular platform, leading to a stronger anti-proliferative ability than heptaplatin itself.

Pillar[n]arene is another popular macrocyclic host in supramolecular chemotherapy. The pillar[n]arene has a high affinity with spermine. Zhang’s group reported that the carboxylated pillar[6]arene-oxaliplatin complex showed a 20% higher anticancer bioactivity than oxaliplatin itself (Figure 16) [91]. The possible mechanism was ascribed to encapsulated oxaliplatin that was released from the carboxylated pillar[6]arene-oxaliplatin complex by competitive replacement with spermine. At the same time, the cytotoxicity toward normal cells was reduced. 

Wang’s group reported bispillar[5]arene-paclitaxel nanoparticles that were spermine and glutathione stimuli-responsive, allowing precisely selective drug release in lung cancer cells that overexpress spermine and glutathione [92]. Peptide-drug conjugates show excellent biocompatibility and tunable morphologies. Wang’s group developed the supramolecular-peptide based on CB[7] and peptide-camptothecin conjugates by non-covalent interactions [93]. When the supramolecular-peptide was internalized into spermine-overexpressed cancer cells, the free peptide-camptothecin conjugates were released by the competitive binding of spermine with CB[7]. As a result, the drastic morphological transformation from nanoparticles to microfibers led to an improved accumulation and retention of supramolecular peptides in tumor cells. The polyamine-responsive “nanoparticle-to-microfiber” transformation exhibited a potential tumor therapy effect. Li’s group synthesized a peptide-pillar[5]arene conjugate as a supramolecular trap, showing a wide spectrum of antitumor activities [94].

#### 3.2.2. Combination of Polyamine Consumption and Photodynamic Therapy (PDT)

Our group explored a new type of PPAB-based photosensitizer (compound **17**), which can cooperatively consume polyamine and produce two photosensitizers through an irreversible chemical reaction [95]. The combination of polyamine consumption and PDT by synergistically destroying tumor cells was a promising polyamine consumption strategy (Figure 17). Compound **17** had lysosomal targeting ability, ratiometric fluorescence imaging capability, polyamine depletion, and enhanced ROS generation. This study provides an irreversible polyamine consumption strategy, which improves anticancer efficacy after being combined with PDT.

## 4. Future Perspectives and Conclusions

Several types of cancer cells (e.g., from prostate cancer, lung cancer, and breast cancer) demonstrate increased intracellular polyamine concentrations, which can be regarded as a biomarker for the timely diagnosis of cancer. Thus, the determination of polyamine contents in biological fluids is increasingly urgent. Many non-optical (GC, HPLC, GC-MS, immunoassays, etc.) and optical (fluorescence or visible color changes) methods have been developed for the quantitative detection of polyamine. Like any analytical method, they have certain limitations and drawbacks, as shown in Table 1. The instrumental, non-optical methods are reliable for polyamine analysis. However, they often necessitate long and tedious sample pretreatment steps and chemical derivatization processes, and require harmful and HPLC-grade quality organic solvents. In particular, the sample pretreatment, including the extraction, enrichment, and derivatization, often needs to be carried out immediately prior to detection [96]. The most common polyamine derivatization approaches include dansyl chloride, benzoyl chloride, *o*-phthalaldehyde, etc. Recently, carrier-mediated membrane-assisted three-phase liquid–liquid extraction coupled with liquid chromatography–mass spectrometry was developed for polyamine determination [97].

Fluorescence detection is known as a low-cost, simple operation, with a high throughput and high sensitivity, but most fluorescent sensors are not completely water-soluble. Some of them show an aggregation-caused quenching effect when they are deposited in the solid, film, or aggregated states. Moreover, the fluorescence signal is susceptible to factors that are unrelated to the analytes, such as instrument parameters, auto-fluorescence, and external environment variations (e.g., temperature and humidity). The development of ratiometric fluorescence signals and AIE-active fluorescence probes would be promising strategies.

Recently, many novel sensing methods and materials have received additional attention. Enzymatic biosensors such as chemiluminescents, electrochemicals, and fluorescence have been developed for polyamine evaluation [98]. Cu^2+^—2-carboxymethylthio-5-mercapto-1,3,4-thiadiazole-functionalized organic thin-film transistors (OTFTs) exhibit the detection and discrimination of multi-polyamines [99]. The electrochemiluminescent (ECL) method was developed for the detection of several polyamines on a microfluidic chip [100]. An ultrahigh-performance supercritical fluid chromatography (UHPSFC) method was utilized for polyamine detection in gentamicin sulfate [101]. The in-situ high-performance chemical receptor-conjugated graphene electronic nose shows potential applications for polyamine detection [102]. Overall, more robust constitution techniques need to be developed to meet the requirements of polyamine determination. In the future, more advanced technology, including intelligent nanofilm biosensors, and multifunctional nanocomposite biosensors, would be a promising research field for polyamine detection.

In addition, medications modulating polyamine levels against biosynthesis, transport and degradation have been investigated as cancer-drug candidates. Since the polyamine compensation mechanism is frequently activated, polyamine biosynthesis and metabolic inhibitors are not ideal candidates. Supramolecular chemotherapy can be applied to treat spermine-overexpressed tumors with excellent targeting and high tumor inhibition with minimal side effects, but they fail to efficiently induce tumor cell apoptosis due to the reversible host–guest interaction between the host and spermine. A combination of polyamine consumption and PDT may present a promising strategy. In general, polyamine-based antitumor treatments still have some way to go before they have a desirable anti-tumor treatment effect. Multiple combination therapeutic approaches based on targeting polyamine depletion would provide new leads in cancer therapeutics.

## Data Availability

Not applicable.

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
