# Peer review of "Recent Advances in Fluorescent Methods for Polyamine Detection and the Polyamine Suppressing Strategy in Tumor Treatment"

_biosensors, 2022, doi:10.3390/bios12080633_

Round 1

Reviewer 1 Report

The manuscript "Recent advances in polyamine detection and polyamine suppressing strategy in tumor" is a review with a double objective: to summarize the recent advances in polyamine detection and to discuss the polyamines suppression strategy.

In the case of the first objective, the manuscript falls very short form being a comprehensive review. It seems to be more like a collection of the authors own works. Both in the Abstract and the Introduction the authors say they will discuss "immunoassays, high performance liquid chromatography (HPLC), chromatography spectrometry [sic]...", none of which are discussed in the manuscript. Moreover, in the few "recent" methods that are indeed described, there is no discussion on the advantages in relation to previous ones.

The second objective is better achieved. However the discussion is still very limited and important examples from the literature are left out.

As general observations, the manuscript is too short for what would be expected for a review of this characteristics. There are grammatical and typographical errors in almost every paragraph, so extensive English proofreading is required.

Reviewer 2 Report

In this review, the authors summarized recent developments in polyamines detection (immunoassays, high performance liquid chromatography (HPLC), chromatography spectrometry, supramolecular sensing system, and chromophore reaction-based fluorescent probes) and tumor polyamines suppression strategies (such as polyamine conjugate, polyamine analogs, combinations target multiple components, spermine responsive supramolecular chemotherapy, combination of polyamine consumption and photodynamic therapy, etc).. I believe that revision is necessary to maintain the publication's high quality.

1.     Authors should show the roadmap for the recent developments in polyamines detection. Author’s may see this work for reference: Nanomaterials, 2022, 12, 2283.

2.     In the caption of Fig. 1, authors should add the full-form of methods used for polyamine detection.

3.     Authors should cite the reference of Figs. 4, 5, 6, 7, 8, 9, 10, 11 and others. 

4.     Authors should discuss the merits and demerits of all discussed methods.

5.     They should add one table and compare the performance of all methods (immunoassays, high performance liquid chromatography (HPLC), chromatography spectrometry, supramolecular sensing system, and chromophore reaction-based fluorescent probes).

6.  They should discuss the Future perspectives and conclusions sections, separately. 

7.     Most of the references are older than 5 years. There are only 2 references of Yr. 2022 are considered. Authors should add more recent references especially of Yr. 2022. 

Round 2

Reviewer 1 Report

Considering the substantial changes introduced in the manuscript I recommend the manuscript to be accepted for publication.

Reviewer 2 Report

The authors have effectively revised the manuscript, and it is now ready for publication.